# Budesonide promotes airway epithelial barrier integrity following double-stranded RNA challenge

Clara Rimmer[1]*, Savas Hetelekides[1], Sophia I. Eliseeva[1], Steve N. Georas[1,2]*, Janelle M. Veazey[2]¤*

**1** Department of Medicine, Pulmonary and Critical Care, University of Rochester, Rochester, New York, United States of America, **2** Department of Microbiology and Immunology, University of Rochester, Rochester, New York, United States of America

¤ Current address: Department of Microbiology and Immunology, Cornell University, Ithaca, New York, United States of America
* clararimmer@gmail.com (CR); Steve_Georas@urmc.rochester.edu (SNG); jv448@cornell.edu (JMV)

**Data Availability Statement:** All relevant data are within the paper and its Supporting information files.

## Abstract

Airway epithelial barrier dysfunction is increasingly recognized as a key feature of asthma and other lung diseases. Respiratory viruses are responsible for a large fraction of asthma exacerbations, and are particularly potent at disrupting epithelial barrier function through pattern recognition receptor engagement leading to tight junction dysfunction. Although different mechanisms of barrier dysfunction have been described, relatively little is known about whether barrier integrity can be promoted to limit disease. Here, we tested three classes of drugs commonly prescribed to treat asthma for their ability to promote barrier function using a cell culture model of virus-induced airway epithelial barrier disruption. Specifically, we studied the corticosteroid budesonide, the long acting beta-agonist formoterol, and the leukotriene receptor antagonist montelukast for their ability to promote barrier integrity of a monolayer of human bronchial epithelial cells (16HBE) before exposure to the viral mimetic double-stranded RNA. Of the three, only budesonide treatment limited transepithelial electrical resistance and small molecule permeability (4 kDa FITC-dextran flux). Next, we used a mouse model of acute dsRNA challenge that induces transient epithelial barrier disruption *in vivo*, and studied the effects budesonide when administered prophylactically or therapeutically. We found that budesonide similarly protected against dsRNA-induced airway barrier disruption in the lung, independently of its effects on airway inflammation. Taken together, these data suggest that an under-appreciated effect of inhaled budesonide is to maintain or promote airway epithelial barrier integrity during respiratory viral infections.

## Introduction

Airway epithelial cells form a physical barrier to the outside world. They are among the first cells to encounter inhaled pathogens, and contribute to airway inflammation by secreting pro-inflammatory cytokines and other mediators [1–3]. Airway epithelial cells are normally tightly

**Funding:** The work reported here was supported by the following grants: R01 HL12424 (SG) from National Institute of Health/ National Heart, Lung, and Blood Institute (https://www.nhlbi.nih.gov/) R01 AI144241 (SG) from National Institute of Health/ National Institute of Allergy and Infectious Disease (https://www.niaid.nih.gov/) F31 HL14079501 (JV) from National Institute of Health/ National Heart, Lung, and Blood Institute (https://www.nhlbi.nih.gov/) T32AI007285 (JV) from National Institute of Health/ National Institute of Allergy and Infectious Disease (https://www.niaid.nih.gov/) The funders had no role in study design, data collection and analysis, decision to publish, or preparation of the manuscript.

**Competing interests:** The authors have declared that no competing interests exist.

connected by adheren junctional proteins that join cells together and tight junctional proteins that promote barrier integrity. Regulation of these junctional protein complexes is critical for maintenance of epithelial barrier integrity, and for epithelial differentiation and activation [3–5]. Epithelial barrier dysfunction is increasingly associated with airway diseases including asthma and COPD, and can be caused by different environmental exposures [1]. Respiratory viruses in particular are notable for their ability to cause marked and sustained perturbation of epithelial barrier integrity [6–10]. The molecular mechanisms by which respiratory viruses disrupt barrier function include inhibiting the assembly of junctional complexes at the cell surface, which render epithelial monolayers "leaky", facilitating paracellular movement of macromolecules in and out of the airway lumen [11]. Synthetic double stranded RNA (dsRNA) engages Toll-like Receptor 3 and can mimic the effects of virus-induced epithelial barrier dysfunction, which has been a useful tool to investigate mechanisms of virally-induced pathogenesis [12,13].

The host inflammatory response must maintain the delicate balance between sufficient potency to clear infection but avoid excessive inflammation that can lead to barrier disruption and tissue injury [14–16]. Inhaled corticosteroids (ICS) such as budesonide, are commonly prescribed to attenuate airway inflammation and lessen airway hyperreactivity [17–20]. ICS suppress the production of pro-inflammatory cytokines and chemokines in asthma. In asthmatic subjects with neutrophilic airway inflammation, potential targets of ICS include the cytokine interleukin-6 (IL-6) and the neutrophil-attracting chemokine CXCL1. In addition to their role in suppressing airway inflammation, ICS might also promote epithelial barrier integrity, but this has not been as well studied in asthma or models of airway inflammation.

Although epithelial barrier dysfunction has been observed in asthma, and is associated with asthma severity, [1,21,22] relatively little is known about the effects of therapeutic compounds in regulating or promoting epithelial barrier integrity. Asthma medications are primarily thought to exert anti-inflammatory effects or act as bronchodilators. For instance, inhaled corticosteroids (ICS) such as budesonide are the mainstay of anti-inflammatory therapy in asthma, and target many cell types in the lung including pro-inflammatory Th2 cells and eosinophils [17,18]. ICS are frequently combined with long-acting beta agonists (LABA) such as formoterol, which have potential anti-inflammatory effects in addition to their bronchodilation properties [20,23]. The leukotriene receptor antagonist montelukast is frequently used to treat subjects with allergic asthma, and works to block the pro-inflammatory effects of leukotrienes [19].

Here we studied the effects of the ICS budesonide, the LABA formoterol, and leukotriene receptor agonist montelukast in a cell culture model of dsRNA-induced barrier dysfunction. We previously demonstrated that double-stranded RNA (polyI:C) is a potent disruptor of barrier integrity in 16HBE cells [12]. Many viruses generate dsRNA during replication and so the dsRNA polyI:C has often been used as simplified model of viral infection [24,25]. We report that budesonide has barrier protective effects, limiting dsRNA-induced paracellular permeability of a model epithelial monolayer. We also used a mouse model of acute dsRNA inhalation that leads to barrier disruption, and studied the effects of inhaled budesonide on barrier function *in vivo*. We report that budesonide protects against dsRNA-induced barrier dysfunction *in vivo*, independently of its effects on airway inflammation.

## Materials and methods

### 16HBE TEER and small molecule flux assay

Monolayers of human 16HBE bronchial epithelial cells passage 17–20 (a gift from Dr. D. C. Gruenert, University of California San Francisco, CA) were grown on Transwell inserts

(Corning; polyester inserts with 0.4 um pores and 0.33 cm$^2$ growth area) then exposed to low-dose poly I:C (0.05 or 0.5 µg/ml, InvivoGen Cat#tlrl-pic; Version#11C21-MM), a synthetic analog of viral double stranded RNA. Barrier function was measured with trans-epithelial electrical resistance (TEER) at 6, 24, and 48 hrs after polyI:C treatment using a voltometer (World Precision Instruments EVOM2). At 48 hrs after treatment, 4 kDa fluorescein isothiocyanate (FITC) dextran (Sigma, used at 10 µg/ml) was applied apically and accumulation of FITC-dextran into the basal chamber was quantified with a Beckman Coulter DTX 880 Multimode fluorescent plate reader 2 hrs later. To assess the effects of selected medications on cell permeability, monolayers were treated with 1–10 µM budesonide (Sigma), montelukast (Sigma), or formoterol (Sigma) 18 hrs prior to polyI:C challenge.

## Western Blot

Monolayers of human 16HBE bronchial epithelial cells passage 17–20 were grown on 6 well plates (Corning), treated with 0.1–1 µM budesonide 18hr prior to addition of low-dose poly I:C (0.5 µg/ml) for an additional 24hrs. Cells were lysed in RIPA buffer containing protease inhibitors (). 40µg protein was loaded onto a 10% SDS-PAGE gel and the gel run at 100V. The protein was transferred to PVDF membranes via TurboBlot semi-dry transfer (BoiRad) and blocked in 5% bovine serum albumin (BSA from Sigma). Primary antibodies were diluted as noted below and incubated with membranes overnight. Secondary antibodies were diluted as noted below and incubated with membrane 1hr. Bands were developed with Clarity Chemiluminescence (BioRad), visualized with BioBlot BXR film (Laboratory Product Sales) and densitometry was assessed with ImageJ.

## Primary antibodies

- Anti-occludin (Invitrogen OC-3F10), 1:1000 dilution in 5% BSA

- Anti-claudin-4 (Invitrogen ZMD.306), 1:1000 dilution in 5% BSA

- Anti-E-cadherin (ThermoFischer 20874-1-AP), 1:1000 dilution in 5% BSA

- Anti-GAPDH (AbCam 8245), 1:50,000 dilution in 5% milk

- Anti-mouse-HRP (GE Healthcare NA934V), 1:10,000 in 5% milk

- Anti-rabbit-HRP (GE Healthcare NA931V), 1:10,000 in 5% milk

## Mice and polyI:C inhalation challenge

Wild-type C57BL/6 were obtained from the National Cancer Institute. All animals were age and sex matched, and treated according to the Institutional Animal Care and Use Committee and Institutional Review Board approval. Mice were administered 10 µg high molecular weight polyI:C (InvivoGen Cat#tlrl-pic; Version#11C21-MM, oropharyngeally (o.p.) daily for three days to induce airway inflammation. This procedure is well-tolerated and results in no overt signs of distress or weight loss. Budesonide (Sigma CAS51333-22-3) was dosed o.p. at 350 or 700 µg/kg for 1–5 days (depending on the experiment). Twenty-four hours after the last instillation, mice were euthanized and the trachea was cannulated. Bronchioalveolar lavage (BAL) was performed with two instillations of 750 µL of phosphate buffered saline (PBS). The type and approximate quantity of cells was analyzed by hemacytometry. The bronchoalveolar lavage fluid was then spun down onto glass slides, stained with hematoxylin and eosin

(FisherBrand 122–911), and leukocytes were quantified. Cell-free supernatants were analyzed for total protein (via Bradford assay), albumin levels (Abcam ab108792 ELISA), and CXCL1 levels (R&D DuoSet DY453 ELISA) according to manufacturer instructions.

### Outside/In leak assay

At 23 hrs post final instillation and 1 hr prior to harvest, 0.2 mg 4kDa FITC-dextran (Sigma#46944) was administered o.p. to mice. BALF was collected as above. Blood was collected via cardiac puncture, and centrifuged for 12 min at 12,000 rpm at 4˚C to separate serum from cellular component. FITC-dextran levels were assessed using a fluorescent plate reader (Beckman Coulter DTX 880 Multimode plate reader).

### Statistical analysis

All values are expressed as means ± 95% standard deviation. Statistical analyses were performed using an unpaired t-test for two groups and ANOVA followed by Tukey's test for multiple groups. A p-value of 0.05 or less was considered statistically significant. All data were analyzed using GraphPad Prism 5.

## Results

### Double-stranded RNA disrupts barrier in 16HBE cells

Before testing the ability of different drug classes to promote barrier integrity, we optimized the dose of polyI:C that transiently disrupts barrier integrity as measured by both TEER and small molecule permeability in 16HBE cells culture. We demonstrate that polyI:C TEER and increases permeability to 4 kDa FITC-dextran in a dose-dependent manner across a monolayer of 16HBE cells (Fig 1A and 1B). We further found that barrier integrity began to recover by 24hrs post-polyI:C treatment with a low dose (0.05μg/ml polyI:C) (Fig 1A). As 0.05μg/ml polyI:C only mildly disrupted barrier, we chose to proceed with 0.5μg/ml polyI:C for our investigation into whether asthma controller medications can prevent barrier disruption and/ or promote more rapid barrier recovery.

### Budesonide, but not formoterol or montelukast promotes barrier integrity in 16HBE cells

We assessed the ability of three asthma medications to prevent/promote barrier recovery following polyI:C-mediated barrier disruption. 16HBE cells were treated with budesonide, formoterol or montelukast 18hrs prior to 0.5μg/ml polyI:C challenge. At concentrations thought to approximate those achieved in epithelial lining fluids *in vivo*, none of the drugs tested had a measurable effect on TEER (Fig 2A). However, the steroid budesonide limited polyI:C-mediated barrier disruption as assayed by small molecule flux (1.983e-6±3.676e-7 vs. 1.429e-6 ±2.504e-7 cm/sec in 0.5μg/ml polyI:C vs. 0.5μg/ml polyIC+0.1 μM budesonide treated cells; p<0.01). The LABA formoterol, and the leukotriene receptor agonist montelukast, failed to mitigate polyI:C-mediated barrier disruption (Fig 2B). To further assess the barrier promoting effect of budesonide, we quantified levels of junctional proteins in 16HBE cells following treatment with polyI:C ± budesonide. Budesonide treatment promoted the maintenance of tight junctional proteins Occludin and Claudin-4 following polyI:C treatment, while the adhesion protein E-cadherin was not altered by either polyI:C or budesonide treatment (Fig 2C).

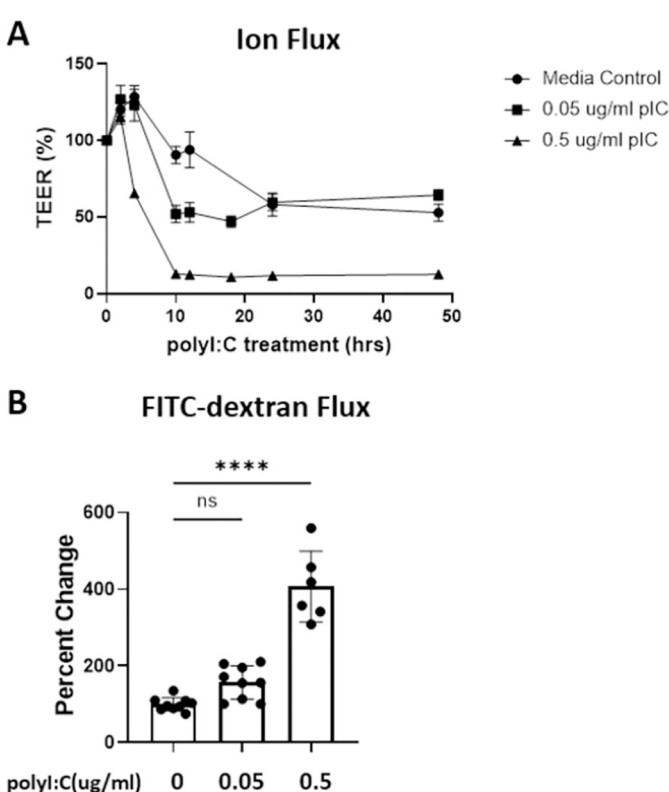

**Fig 1. Double-stranded RNA (polyI:C) disrupts barrier integrity of 16HBE cells in dose-dependent manner.**
16HBE cells were grown to confluency (over 800 ohm) and treated with either vehicle, 0.5, or 0.05 μg/ml polyI:C. A)
TEER was monitored at 6, 24, 30, and 48hrs post polyI:C addition. B) At 48hrs post polyI:C, 10 μg/ml 4kDa FITC-
dextran was applied apically and the amount of FITC-dextran translocation to the basal chamber was quantified 2hrs
later using a fluorescent plate reader. Data are mean ± standard deviation. One way ANOVA followed by unpaired
Tukey's multiple comparisons test. **** p<0.0001.

## Budesonide limits outside/in leak, but not inside/out leak *in vivo*

We next asked if budesonide could similarly promote barrier integrity *in vivo*. Pre-treatment
of mice with 700 μg/kg inhaled budesonide attenuated polyI:C-mediated outside/in barrier
break as seen by less 4kDa FITC-dextran leak out of the airspace and into serum (59.0±9.6 vs.
80.4±14.0 percent change over vehicle-treated in BAL of polyI:C vs. polyIC+700 μg budeso-
nide treated mice; p<0.01) (Fig 3A and 3B). However, budesonide did not limit inside/out
leak as the amount of total protein and albumin in the BAL fluid was not significantly changed
with budesonide treatment (Fig 3C and 3D).

## Budesonide does not limit neutrophil accumulation or pro-inflammatory cytokines *in vivo*

Mice challenged with inhaled polyI:C also developed neutrophilic airway inflammation, as
previously reported [13]. Budesonide treatment did not alter levels of the neutrophil chemoat-
tractant CXCL1 (Fig 4A), but levels of IL-6 and interferon-lambda were reduced with budeso-
nide pre-treatment (Fig 4B and 4C). Furthermore, budesonide did not attenuate polyI:C-
induced neutrophil recruitment, but in fact enhanced neutrophil accumulation at the highest
dose used (Fig 4D).

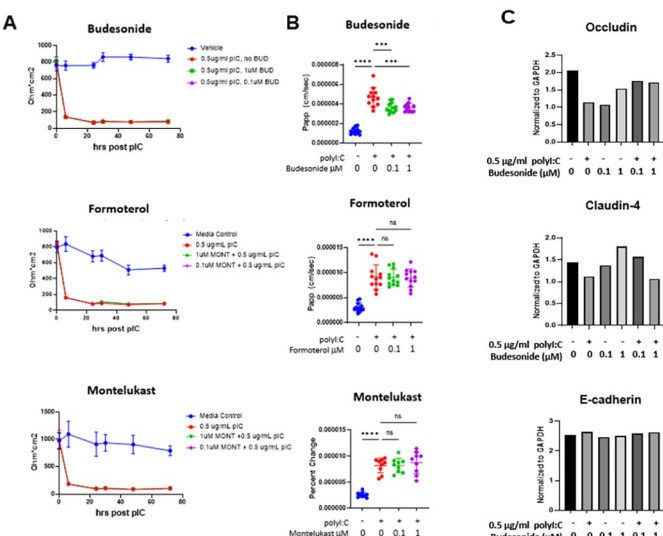

**Fig 2. Budesonide, formoterol or montelukast do not impact TEER, but budesonide attenuates small molecule flux.** 16HBE cells were grown to confluency (over 800 ohm) and treated with vehicle or 0.5 μg/ml polyI:C and 0.1–1μM drug. A) TEER was monitored at 6, 24, 30, 48, and 72hrs post polyI:C addition. B) At 48hrs post polyI:C, 10 μg/ml 4kDa FITC-dextran was applied apically and the amount of FITC-dextran translocation to the basal chamber was quantified 2hrs later using a fluorescent plate reader. C) 16HBE cells were treated as in A and B, but lysed in RIPA buffer for protein analysis by Western Blot. Band intensity was quantified with ImageJ and values normalized to the loading control GAPDH. Data are mean ± standard deviation. One way ANOVA followed by unpaired Tukey's multiple comparisons test. ***p<0.01, **** p<0.0001.

## Budesonide given at the time of polyI:C challenge limits barrier disruption *in vivo*

We next interrogated the therapeutic potential of budesonide given after polyI:C challenge to mitigate barrier disruption *in vivo*. Budesonide given the same day as (Group A), or after (Groups B-C), polyI:C inhalation attenuated outside/in leak, as seen by less FITC-dextran leak out of the airspace (Fig 5A and 5B). Same day budesonide similarly limited inside/out leak (Fig 5C and 5D). Finally, therapeutic dosing of budesonide (administered after polyI:C challenge) did not attenuate neutrophil accumulation or CXCL1 levels following polyI:C inhalation, consistent with finding from prophylactic dosing (Fig 5E and 5F).

## Discussion

While the anti-inflammatory effects of asthma medications are well known, less is known about the ability of these medications to directly promote epithelial barrier integrity. Using dsRNA as a model of acute inflammation and barrier disruption, we show that budesonide, but not formoterol or montelukast, is able to promote barrier integrity in both a human bronchial epithelial cell monolayer, as well as an *in vivo* mouse model. Specifically, budesonide given before polyI:C challenge (mimicking long-term asthma controller medication) in 16HBE cells, or in an *in vivo* mouse model, limits the degree of barrier break (reduced FITC-dextran flux *in vitro*, as well as reduced outside/in leak *in vivo*). Importantly, budesonide treatment did not attenuate neutrophil accumulation or inside/out leak *in vivo*- suggesting normal ability to control pathogens remains un-impaired when barrier is enhanced. Further study is needed to determine the mechanism by which budesonide promotes airway epithelial barrier integrity.

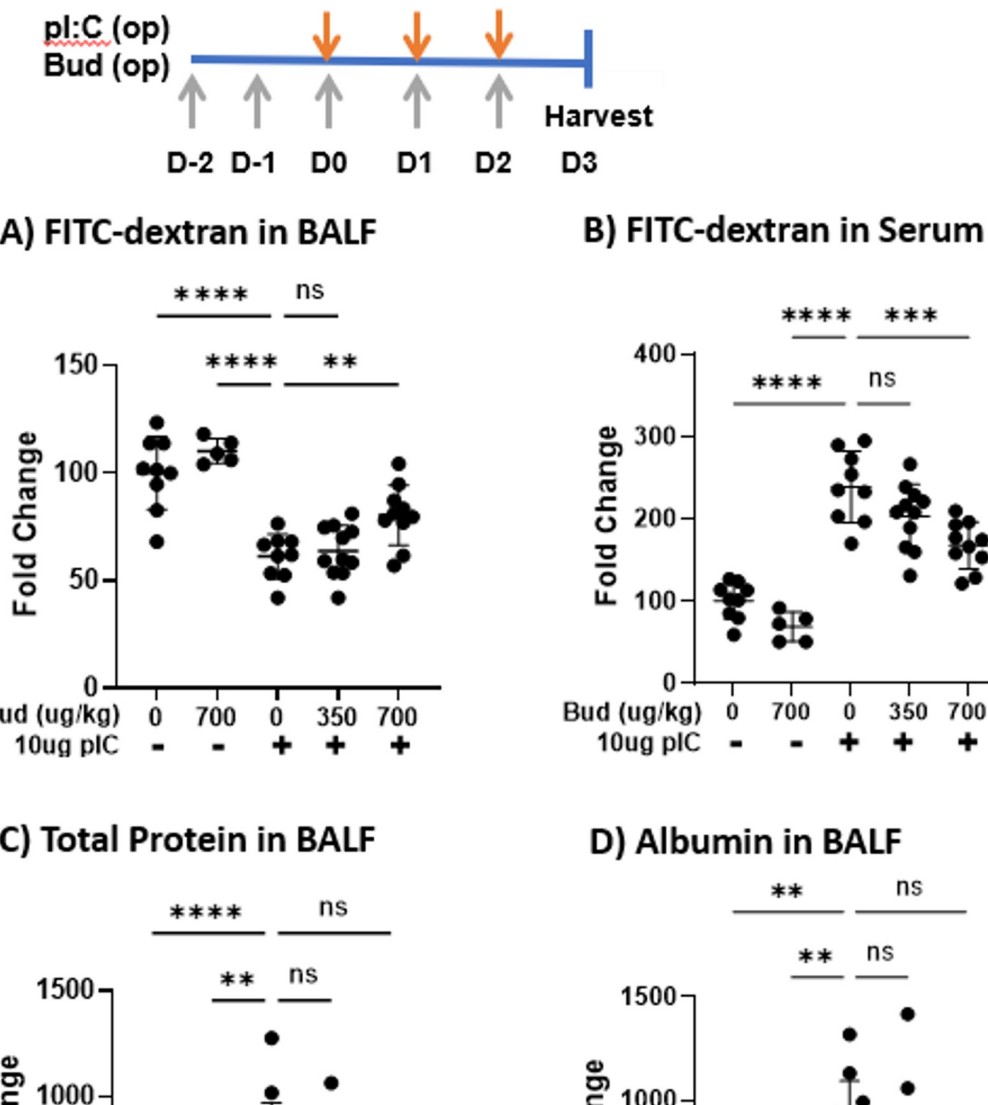

**Fig 3. Budesonide treatment promotes outside/in barrier integrity *in vivo*.** C57B/6 mice were given vehicle or 350–700 μg/kg budesonide (Bud) oropharyngeally (o.p.) on days -2-2, and 10 μg polyI:C o.p. on days 0–2. On day 3, 0.2 mg 4kDa FITC-dextran was instilled o.p. and 1hr later BALF and blood were collected. A-B) BALF and serum were analyzed for FITC-dextran levels using a fluorescent plate reader. C-D) Total protein and albumin levels in BALF were assessed via Bradford and ELISA respectively. Data are mean ± standard deviation. One way ANOVA followed by unpaired Tukey's multiple comparisons test. ** $p < 0.01$; ***$p < 0.001$; **** $p < 0.0001$.

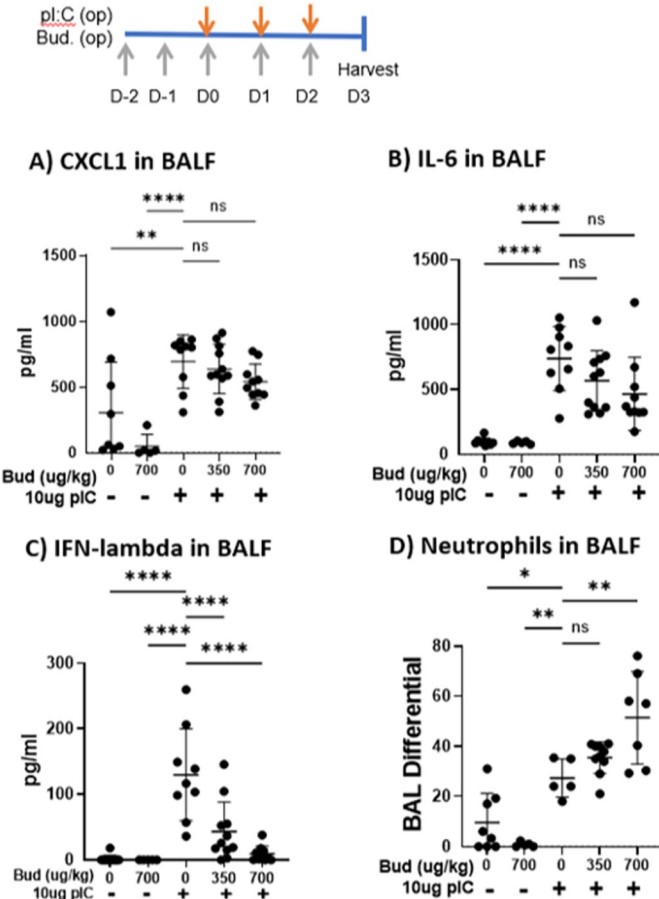

**Fig 4. Budesonide treatment does not attenuate neutrophils and reduces IL-6 and interferon levels.** C57B/6 mice were given vehicle or 350–700 μg/kg budesonide oropharyngeally (o.p.) on days -2-2, and 10 μg polyI:C o.p. on days 0–2. On day 3, 0.2 mg 4kDa FITC-dextran was instilled o.p. and 1 hr later BALF was harvested. A-C) CXCL1, IFN-lambda and IL-6 levels were quantified via ELISA, and D) neutrophils were assessed via cytospin and hematoxylin and eosin staining. Data are mean ± standard deviation. One way ANOVA followed by unpaired Tukey's multiple comparisons test. *p<0.05, ** p< 0.01; ***p<0.001; **** p<0.0001.

Inhaled corticosteroids are the mainstay of treatment for asthma, and exert multiple beneficial anti-inflammatory and immunomodulatory effects to counter allergic airway inflammation [26]. As these medications are often taken long-term as maintenance therapies, we investigated treatment with budesonide, formoterol or montelukast administered prior to polyI:C challenge. With this model we found budesonide significantly enhances outside/in barrier integrity (Fig 3), suggesting that one of the beneficial effects of maintenance inhaled corticosteroids might be their ability to promote epithelial barrier integrity following respiratory viral infections. Airway barrier function is challenging to study in human subjects, but future studies using non-invasive approaches will provide new insights into how asthma therapeutics impact this important aspect of airway dysfunction in asthma.

Our results confirm and extend the results of Heijink et al who found that 16 nM budesonide attenuated polyI:C-induced airway epithelial barrier dysfunction *in vitro*, as determined by electrical cell-surface impedance sensing [27]. Importantly we show that a lower dose of budesonide protects against polyI:C-induced increase in epithelial permeability to small molecule flux (Fig 2). Prophylactic dosing with inhaled budesonide *in vivo* also significantly

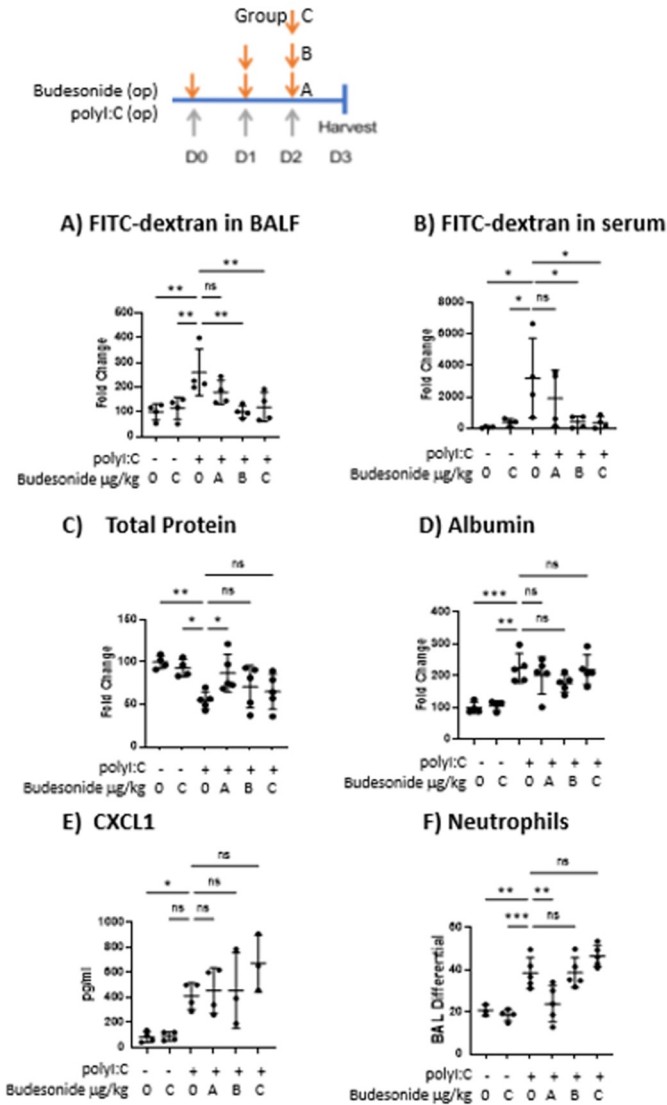

**Fig 5. Therapeutic budesonide treatment promotes barrier integrity without hampering neutrophil accumulation.** C57B/6 mice were given 10 μg polyI:C o.p. on days 0–2. On either days 0–2 (group A), days 1–2 (group B) or day 2 (group C), mice were also given vehicle or 700 μg/kg budesonide o.p.. On D3, 0.2 mg 4kDa FITC-dextran was instilled o.p. and 1hr later BALF and blood were harvested. A-B) Total protein and albumin levels in BALF were assessed via Bradford and ELISA respectively. C-D) FITC-dextran levels in BALF and serum were analyzed using a fluorescent plate reader. E-F) Neutrophils were assessed via cytospin and H&E staining, and CXCL1 was quantified via ELISA. Data are mean ± standard deviation. One way ANOVA followed by unpaired Tukey's multiple comparisons test. $^*$ p $<$0.05, $^{**}$p$<$0.01, $^*$p$<$0.001.

attenuated polyI:C-induced barrier disruption, as reflected by reduced translocation of inhaled FITC-dextran out of the airspaces and into serum (outside/in leak, Fig 3). Interestingly, this barrier protective effect of budesonide was apparent despite an increase recovery of BAL neutrophils from budesonide-treated mice. Increased neutrophils and no change in BAL CXCL1 levels, are consistent with the known effects of glucocorticoids in potentiating neutrophil survival [28] and argue that the pro-barrier integrity effects of budesonide on the epithelial barrier *in vivo* are not simply due to overall reduced levels of airway inflammation. Taken together,

we conclude that budesonide promotes barrier integrity without hindering normal leukocyte accumulation to the site of injury or promoting inflammation.

We observed that prophylactic budesonide attenuated the translocation of FITC-dextran out of the airspaces and into serum (outside/in leak), without significantly inhibiting BAL protein or albumin levels (inside/out leak). This difference in outside/in vs. inside/out leak highlights that small molecule flux is not a simple opening of a hole in the barrier, but is a highly regulated process with directionality. One potential caveat to these findings is that total protein/albumin leak into the BALF quantifies what has leaked over the 4 days of treatment, whereas the FITC-dextran leak out of BALF measures leak over only 1hr. It is therefore possible that budesonide treatment attenuates acute leak but has less effect on long-term leak. This seems unlikely as the last budesonide treatment is 24 hrs prior to FITC-dextran administration thereby minimizing any "acute" effects of the drug at the time of FITC-dextran administration. Furthermore, budesonide given after polyI:C inhalation attenuated inside/out leak as well as outside/in leak (Fig 5), indicating budesonide is capable of limiting inside/out barrier integrity in specific contexts. Future work is needed to understand the mechanism by which budesonide promotes inside/out barrier integrity when given after, but not prior to, polyI:C challenge.

Ideally, promoting barrier integrity would not come at the expense of hindering immune cell infiltration during an infection. Budesonide is well known as a steroid that does not increase risk of severe viral infections [18]. In alignment with this, here we report budesonide treatment did not prevent neutrophil accumulation into the airspace following polyI:C challenge, and may even promote leukocyte accumulation (Fig 4). However, the boost in immune cells is not accompanied by elevated pro-inflammatory cytokines as budesonide treatment did not lower CXCL1 levels. Therefore, budesonide promotes barrier integrity without hindering normal leukocyte accumulation to the site of injury or promoting inflammation. Future work aims to extend these studies by investigating the mechanism by which budesonide promotes barrier integrity during virally-induced asthma exacerbations.

## Supporting information

**S1 File. Detailed protocol information.**
(DOCX)

**S2 File. Uncropped Western Blots.**
(PDF)

**S1 Rawdata. Raw data files.**
(XLSX)

## Acknowledgments

We acknowledge Dr. Patrick Donohue, Kristie Stiles and Traci Pressley for their help with conducting experiments.

## Author Contributions

**Conceptualization:** Clara Rimmer, Savas Hetelekides, Steve N. Georas, Janelle M. Veazey.

**Data curation:** Clara Rimmer, Savas Hetelekides, Sophia I. Eliseeva, Janelle M. Veazey.

**Formal analysis:** Savas Hetelekides, Steve N. Georas, Janelle M. Veazey.

**Funding acquisition:** Steve N. Georas, Janelle M. Veazey.

**Investigation:** Clara Rimmer, Savas Hetelekides, Sophia I. Eliseeva, Janelle M. Veazey.

**Methodology:** Clara Rimmer, Savas Hetelekides, Sophia I. Eliseeva, Janelle M. Veazey.

**Project administration:** Steve N. Georas, Janelle M. Veazey.

**Resources:** Steve N. Georas.

**Supervision:** Steve N. Georas, Janelle M. Veazey.

**Validation:** Clara Rimmer, Savas Hetelekides, Steve N. Georas, Janelle M. Veazey.

**Visualization:** Janelle M. Veazey.

**Writing – original draft:** Steve N. Georas, Janelle M. Veazey.

**Writing – review & editing:** Steve N. Georas, Janelle M. Veazey.

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
