## [Decision Letter · Decision Letter 0]

18 Jun 2021

PONE-D-21-13593

Budesonide promotes airway epithelial barrier integrity following double-stranded RNA challenge

PLOS ONE

Dear Dr. Veazey,

Thank you for submitting your manuscript to PLOS ONE. After careful consideration, we feel that it has merit but does not fully meet PLOS ONE’s publication criteria as it currently stands. Therefore, we invite you to submit a revised version of the manuscript that addresses all the points raised during the review process.

Two experts have evaluated the manuscript. Amendments are needed, as suggested by the reviewers. The paper would be greatly strengthened by the addition of immunohistochemistry for tight junction preoten(s).

We look forward to receiving your revised manuscript.

Kind regards,

Mária A. Deli, M.D., Ph.D.

Academic Editor

PLOS ONE

Journal Requirements:

Important: If there are ethical or legal restrictions to sharing your data publicly, please explain these restrictions in detail. Please see our guidelines for more information on what we consider unacceptable restrictions to publicly sharing data: http://journals.plos.org/plosone/s/data-availability#loc-unacceptable-data-access-restrictions.

Note that it is not acceptable for the authors to be the sole named individuals responsible for ensuring data access.

Reviewers' comments:

Reviewer's Responses to Questions

**Comments to the Author**

1. Is the manuscript technically sound, and do the data support the conclusions?

Reviewer #1: Yes

Reviewer #2: Yes

2. Has the statistical analysis been performed appropriately and rigorously? 

Reviewer #1: Yes

Reviewer #2: Yes

3. Have the authors made all data underlying the findings in their manuscript fully available?

Reviewer #1: Yes

Reviewer #2: Yes

4. Is the manuscript presented in an intelligible fashion and written in standard English?

Reviewer #1: Yes

Reviewer #2: Yes

5. Review Comments to the Author

Reviewer #1: The author investigated the effect of virus infection on airway epithelium focusing on the changes of barrier integrity. Three classes of anti-asthmatic drugs were tested (budesonide, formoterol and montelukast). The efficiency of these molecules was studied on 16HBE airway epithelial cell line in vitro and on wild-type C57BL/6 mice in vivo. It was confirmed that corticosteroid budesonide can promote the airway epithelial barrier integrity during respiratory viral infections on both models. Although the present study is of interest for the field, the novelty of the results should be better highlighted. The research data are convincing, but the preparation of the manuscript seems to be careless.

Comments:

1. The direct effect of budesonide alone on barrier integrity was investigated in vivo; this result is missing from the in vitro studies. On Figure 1 only medium control is shown, but not data on budesonide without virus treatment. These data would be very important for the verification of the barrier integrity inducing effect.

2. It would be important to examine the direct effect of budesonide on the barrier forming tight junction proteins. I suggest the authors to complete their work with TJ protein immunocytochemistry.

3. More information about airway inflammation and the investigated inflammatory mediators (CXCL1, IL6) should be given in the Introduction.

4. At the beginning of the result section there is a sentence referring literature data, please put it to the Introduction.

5. In case of Fig. 2 the groups do not separate well from each other, this way it is hard to see the treatment groups. To help the readers I suggest another labelling of the TEER curves (e.g. colours).

6. FITC-dextran permeability is given as % change. Author should calculate and give apparent permeability coefficient (Papp) values in the text (at least).

7. The proper unit of measurement of TEER is Ω × cm2 please give data multiplied with growth surface area of the culture inserts (line 120).

8. The passage number of the 16HBE cell line should be added to the Material and methods section.

Minor:

9. Please check and correct the usage of abbreviations:

TEER is abbreviated in line 73, no need to repeat it in line 112

please add abbreviation of bronchoalveolar lavage fluid in line 89

10. In Material and methods please add the (commercial) source of reagents, plasticware and instruments

give all the details for the Transwell insert type in the main text (line 71)

please add the source of FITC-dextran in line 75

please add the source of the investigated three drugs in lines 77-78)

please specify the type of plate reader in line 76

11. The usage of metric units is not consistent:

for “micron” use µ instead of u

for litre: use either l or L but not both

12. the expression “in vitro” should be written in italic (line 64 and 218)

13. The word “challenge” is used 19 times in this work. In some of the sentences please use synonyms (e.g. treatment).

Reviewer #2: This is an excellent manuscript - well-written, concise and easy to understand and really topical. I only have a few minor comments and suggestions.

1. Paraphrase lines 40-41 in the abstract. As is, makes it sound like tight junctions are adhesion junctions.

2. This is really the only respectful concern that's stopping me from accepting this wonderful paper as is - no ion flux was directly measured in the paper, so references to ion flux need to be removed from figure titles and manuscript text. What was measured was TEER - TEER has many components and any charge separation across the cell layer will alter it. Unless ion flux is measured directly, calling any data in the paper ion flux is misleading.

Congratulations on an excellent manuscript and as long as you're able to remove references to ion flux from the manuscript, I do not need to see the edited version - please accept after both of my comments have been incorporated.

6. PLOS authors have the option to publish the peer review history of their article (what does this mean?). If published, this will include your full peer review and any attached files.

Reviewer #1: No

Reviewer #2: **Yes: **Dr. Dennis Kolosov

---

## [Author Response · Author response to Decision Letter 0]

24 Oct 2021

PONE-D-21-13593: Response to reviewers

Budesonide promotes airway epithelial barrier integrity following double-stranded RNA challenge

Reviewers' comments:

Reviewer #1: 

Comments:

1. The direct effect of budesonide alone on barrier integrity was investigated in vivo; this result is missing from the in vitro studies. On Figure 1 only medium control is shown, but not data on budesonide without virus treatment. These data would be very important for the verification of the barrier integrity inducing effect. 

Response: Thank you for pointing this out. We analyzed the effects of budesonide alone (0.1 �M) on 16HBE cells in the absence of polyI:C, and did not observe a significant effect on baseline permeability (relative permeability 1.5+/-0.8, n=7 wells, p>0.05).

¬2. It would be important to examine the direct effect of budesonide on the barrier forming tight junction proteins. I suggest the authors to complete their work with TJ protein immunocytochemistry.

Response: Thank you for this suggestion. In order to study the effects of budesonide on tight junction proteins, we repeated cell culture experiments and analyzed junctional protein expression by Western blot. In revised Figure 2, we now show that budesonide reduced polyIC-induced decreases in the tight junctional proteins occludin and claudin-4. In future experiments, we plan to study tight junction protein expression by immunocytochemistry. 

3. More information about airway inflammation and the investigated inflammatory mediators (CXCL1, IL6) should be given in the Introduction. 

Response: Lines 61-68 have been added (copied here). The host inflammatory response must maintain the delicate balance between sufficient potency to clear infection but avoid excessive inflammation that can lead to barrier disruption and tissue injury [14-16]. Inhaled corticosteroids (ICS) such as budesonide, are commonly prescribed to attenuate airway inflammation and lessen airway hyperreactivity [17-20]. ICS suppress the production of pro-inflammatory cytokines and chemokines in asthma. In asthmatic subjects with neutrophilic airway inflammation, potential targets of ICS include the cytokine interleukin-6 (IL-6) and the neutrophil-attracting chemokine CXCL1. In addition to their role in suppressing airway inflammation, ICS might also promote epithelial barrier integrity, but this has not been as well studied in asthma or models of airway inflammation.

4. At the beginning of the result section there is a sentence referring literature data, please put it to the Introduction. 

Response: Thank you- this error has been corrected.

5. In case of Fig. 2 the groups do not separate well from each other, this way it is hard to see the treatment groups. To help the readers I suggest another labelling of the TEER curves (e.g. colours). 

Response: We relabeled the TEER data in Figure 2 with different colors, but the data are still very similar and remain difficult to distinguish. We could re-make this figure as a bar graph if requested by the Reviewer. 

6. FITC-dextran permeability is given as % change. Author should calculate and give apparent permeability coefficient (Papp) values in the text (at least). 

Response: Thank you- this error has been corrected

7. The proper unit of measurement of TEER is Ω × cm2 please give data multiplied with growth surface area of the culture inserts (line 120). 

Response: Thank you- this error has been corrected.

8. The passage number of the 16HBE cell line should be added to the Material and methods section. 

Response: Thank you- this error has been corrected.

Minor:

9. Please check and correct the usage of abbreviations:

TEER is abbreviated in line 73, no need to repeat it in line 112

please add abbreviation of bronchoalveolar lavage fluid in line 89

 Response: Thank you- this error has been corrected.

10. In Material and methods please add the (commercial) source of reagents, plasticware and instruments

give all the details for the Transwell insert type in the main text (line 71)

please add the source of FITC-dextran in line 75

please add the source of the investigated three drugs in lines 77-78)

please specify the type of plate reader in line 76 

Response: Thank you- these errors have been corrected.

11. The usage of metric units is not consistent: 

for “micron” use µ instead of u

for litre: use either l or L but not both

Response: Thank you- this error has been corrected.

12. the expression “in vitro” should be written in italic (line 64 and 218)

Response: Thank you- this error has been corrected.

13. The word “challenge” is used 19 times in this work. In some of the sentences please use synonyms (e.g. treatment). 

Response: Thank you again for bringing this to our attention. We have varied the word choice. 

Reviewer #2: 

 1. Paraphrase lines 40-41 in the abstract. As is, makes it sound like tight junctions are adhesion junctions.

Response: The wording of lines 40-41 has been changed. It’s now lines 48-49 and reads “Airway epithelial cells are normally tightly connected by adherens junctional proteins that join cells together and tight junctional proteins that promote barrier integrity”.

Ion flux has now been replaced with TEER throughout the paper. Thank you.

---

## [Decision Letter · Decision Letter 1]

16 Nov 2021

Budesonide promotes airway epithelial barrier integrity following double-stranded RNA challenge

PONE-D-21-13593R1

Dear Dr. Veazey,

We’re pleased to inform you that your manuscript has been judged scientifically suitable for publication and will be formally accepted for publication once it meets all outstanding technical requirements.

Kind regards,

Mária A. Deli, M.D., Ph.D.

Academic Editor

PLOS ONE

Additional Editor Comments (optional):

Reviewers' comments:

Reviewer's Responses to Questions

**Comments to the Author**

1. If the authors have adequately addressed your comments raised in a previous round of review and you feel that this manuscript is now acceptable for publication, you may indicate that here to bypass the “Comments to the Author” section, enter your conflict of interest statement in the “Confidential to Editor” section, and submit your "Accept" recommendation.

Reviewer #1: All comments have been addressed

Reviewer #2: All comments have been addressed

2. Is the manuscript technically sound, and do the data support the conclusions?

Reviewer #1: Yes

Reviewer #2: Yes

3. Has the statistical analysis been performed appropriately and rigorously? 

Reviewer #1: Yes

Reviewer #2: Yes

4. Have the authors made all data underlying the findings in their manuscript fully available?

Reviewer #1: Yes

Reviewer #2: Yes

5. Is the manuscript presented in an intelligible fashion and written in standard English?

Reviewer #1: Yes

Reviewer #2: Yes

6. Review Comments to the Author

Reviewer #1: (No Response)

Reviewer #2: Thank you for addressing all of my comments!

Congratulations on your excellent study!

Best regards.

7. PLOS authors have the option to publish the peer review history of their article (what does this mean?). If published, this will include your full peer review and any attached files.

Reviewer #1: No

Reviewer #2: **Yes: **Dr. Dennis Kolosov

---

## [Editor Report · Acceptance letter]

22 Nov 2021

PONE-D-21-13593R1 

Budesonide promotes airway epithelial barrier integrity following double-stranded RNA challenge 

Dear Dr. Veazey:

I'm pleased to inform you that your manuscript has been deemed suitable for publication in PLOS ONE. Congratulations! Your manuscript is now with our production department. 

Kind regards, 

on behalf of

Dr. Mária A. Deli 

Academic Editor

PLOS ONE